**Data Availability Statement:** The cross-sectional NCD data underlying this analysis are available for

# Waist circumference and glycaemia are strong predictors of progression to diabetes in individuals with prediabetes in sub-Saharan Africa: 4-year prospective cohort study in Malawi

**Wisdom P. Nakanga**[1,2]*, **Amelia C. Crampin**[2], **Joseph Mkandawire**[2], **Louis Banda**[2], **Rob C. Andrews**[1], **Andrew T. Hattersley**[1], **Moffat J. Nyirenda**[3,4], **Lauren R. Rodgers**[5]

1 National Institute for Health Research (NIHR), Exeter Clinical Research Facility, University of Exeter, Exeter, United Kingdom, 2 Malawi Epidemiology and Intervention Research Unit (MEIRU), Karonga and Lilongwe, Malawi, 3 London School of Hygiene and Tropical Medicine (LSHTM), London, United Kingdom, 4 Medical Research Council/ Uganda Virus and Research Institute and LSHTM Uganda Research Unit, Entebbe, Uganda, 5 Institute of Health Research, University of Exeter Medical School, Exeter, United Kingdom

* wn236@exeter.ac.uk

## Abstract

Sub-Saharan Africa is projected to have the highest increase in the number of people with diabetes worldwide. However, the drivers of diabetes in this region have not been clearly elucidated. The aim of this study was to evaluate the incidence of diabetes and the predictors of progression in a population-based cohort with impaired fasting glucose (IFG) in Malawi. We used data from an extensive rural and urban non-communicable disease survey. One hundred seventy-five, of 389 individuals with impaired fasting glucose (IFG) at baseline, age 48 ±15 years and body mass index 27.5 ±5.9 kg/m2 were followed up for a median of 4.2 years (714 person-years). Incidence rates were calculated, and predictors of progression to diabetes were analysed using multivariable logistic regression models, with overall performance determined using receiver operator characteristics (ROC) curves. The median follow-up was 4.2 (IQR 3.4–4.7) years. Forty-five out of 175 (26%) progressed to diabetes. Incidence rates of diabetes were 62.9 per 1000 person-years 95% CI, 47.0–84.3. The predictors of progression were higher; age (odds ratio [OR] 1.48, P = 0.046), BMI (OR 1.98, P = 0.001), waist circumference (OR 2.50, P<0.001), waist-hip ratio (OR 1.40, P = 0.03), systolic blood pressure (OR 1.56, P = 0.01), fasting plasma glucose (OR 1.53, P = 0.01), cholesterol (OR 1.44, P = 0.05) and low-density lipoprotein cholesterol (OR 1.80, P = 0.002). A simple model combining fasting plasma glucose and waist circumference was predictive of progression to diabetes (ROC area under the curve = 0.79). The incidence of diabetes in people with IFG is high in Malawi and predictors of progression are like those seen in other populations. Our data also suggests that a simple chart with probabilities of progression to diabetes based on waist circumference and fasting plasma glucose could be used to identify those at risk of progression in clinical settings in sub-Saharan Africa.

download via the LSHTM Data Compass Repository (https://datacompass.lshtm.ac.uk/cgi/request_doc?docid=6714). However, owing to data protection concerns, there are restrictions on access to this underlying data. Given their linkages to the longitudinal data produced by MEIRU, these data require anonymization using statistical disclosure control techniques to make them sufficiently safe for direct access. The anonymization process is scheduled for the last quarter of 2022. In the meantime, data access to bona fide researchers for specific research purposes is possible through completing a data request form on the LSHTM data compass repository. Once the anonymization process has been completed, direct access will be granted via the same repository.

**Funding:** This work was supported by the UK Medical Research Council (MRC) and the UK Department for International Development (DFID) under the MRC/DFID Concordat agreement through a strategic award to MJN (Project Reference: MC_UP_1204/16). The funders had no role in study design, data collection and analysis, decision to publish, or manuscript preparation.

**Competing interests:** The authors have declared that no competing interests exist.

**Abbreviations:** BMI, Body Mass Index; FPG, Fasting plasma glucose; HDL, High-density lipoprotein; IFG, Impaired Fasting Glucose; LDL, Low-density lipoprotein; NGT, Normal Glucose Tolerance; SSA, Sub Saharan Africa.

# Introduction

Sub Saharan African (SSA) will face the largest increase in diabetes worldwide unless drastic interventions are implemented. Presently 19 million people in the region have diabetes, and this is projected, by the International Diabetes Federation (IDF), to rise to 34.2 million people, an increase of 143%, by 2045 [1]. This rise is expected to correspond with a similar surge in the number of people with prediabetes, a transition stage between normal glucose tolerance and diabetes. Individuals with prediabetes have a higher risk of developing diabetes and macrovascular disease than those with normal glucose tolerance and therefore offer a unique target population for identification and intervention [2–5].

Few studies have determined the incidence rate and the modifiable risk factors that predispose to the development of diabetes amongst the population with prediabetes in SSA, with no studies having being conducted in Malawi [6]. The factors that determine the progression of prediabetes have been widely studied in high-income countries and include positive family history, obesity and age [7, 8]. But these factors cannot be generalised to SSA, where patients with diabetes are younger and thinner than in other regions [9, 10]. Moreover, the rate of progression in prediabetes individuals has been shown to vary between different study populations [11, 12]. In SSA the rate of progression may further be influenced by the distinct diabetes phenotypes in addition to early environmental exposures to malnutrition and infections.

The diagnosis of prediabetes can be based on measurement of fasting glucose, oral glucose tolerance test (OGTT) or HbA1c. However, in a setting where the health systems are constrained with limited health workers and a lack of medical supplies, OGTT is not practical and are seldom done in SSA. HbA1c is also not frequently used because of cost limitations, high prevalence of haemoglobin variants and anaemia [13, 14]. Fasting plasma glucose (FPG) is cheaper, easy to measure, and frequently used in clinical practice for diagnosing and monitoring diabetes in SSA. Prediabetes diagnosed solely based on fasting glucose is specifically called impaired fasting glucose (IFG) and is defined by the World Health Organization and the American Diabetes Association using different cut-offs.

In this study, we attempt to determine the incidence rate of diabetes in people with IFG and to assess the predictors of progression in the follow-up cohort of an epidemiological survey, conducted in a representative urban and rural population in Malawi.

# Methods

## Study population

The population for this study was a subset of participants with IFG who took part in the Malawi Epidemiology and Intervention Research Unit Non-Communicable Disease Survey. This cross-sectional population study was conducted on a representative sample of 13,878 adults (>18 years) participants from a rural district, Karonga, and 15,013 from the urban city, Lilongwe, between 2013 and 2016. The methodology and results of the baseline cross-sectional study have been published elsewhere [15, 16]. From the total of 28,891 participants, all the participants with IFG according to the WHO classification an FPG greater or equal to 6.1 mmol/L and less than seven mmol/L with no history of diabetes diagnosis and not taking diabetes medication (n = 389) were followed-up.

We used the Cochran formula to find the margin of error in our estimate of diabetes incidence compared to the actual value. Assuming a prevalence of 4% conversion to diabetes per year, 389 subjects would give us a 1.1% absolute precision for a 95% confidence level. As only 175 subjects could be recruited, we could estimate the prevalence of diabetes conversion of 4% per year with a margin of error of 2.1% based on the width of the 95% confidence intervals.

### Ethics statement

All activities were approved by the local ethics committee (protocol no. 17/12/1953, National Health Sciences Research Committee, Lilongwe. Malawi) and complied with the Declaration of Helsinki. Written, informed consent was obtained from all the participants before recruitment into the study.

### Measures

**Demographic, education and health-related behaviours.** A trained field team visited the participant at home. Demographic, socioeconomic status, medical and family history, tobacco and alcohol use were captured using a structured interviewer-administered questionnaire. Family history was considered "positive" if immediate family (parents or siblings) were reported to have been diagnosed with diabetes. We generated proxy wealth scores using locally determined monetary values of household assets, categorised into thirds across the total baseline population [17]. We combined self-reported physical exercise duration (minutes) and intensity (pre-coded activities, grouped into high exertion, low exertion, or sedentary) in the previous week (both at work and during leisure time) to generate average metabolic equivalent of task (MET) data per day, and categorised these into whether or not the WHO recommendations of at least 600 Total Physical Activity MET minutes per week were met [18].

**Anthropometric measurements.** Height, weight, waist circumference and hip circumference were measured twice, in light clothing and without shoes, using calibrated Seca scales, stadiometers, and flexible tape measures. The mean of the two measurements was used in the analyses. BMI was calculated by dividing weight (kg) by height squared ($m^2$). Three seated blood pressure measurements, with 5 min rest in between, were collected on the right arm using portable sphygmomanometers (OMRON-Healthcare-CoHEN-7211; Kyoto, Japan). The mean of the last two blood pressure readings was used in the analysis. The same protocol was used to take these measures at follow-up.

**Biochemical analyses.** Fasting blood samples were drawn in the morning (from 0500 h) after a minimum 8 h fast by a trained nurse who also provided HIV screening using a rapid diagnostic test according to standard operating procedures. The samples were transported, on ice, to the onsite project laboratory for processing (mean time between collection and processing 2.6 hours). Samples for glucose measurement were collected in sodium fluoride (NaF) tubes. We used the hexokinase glucose-6-phosphate dehydrogenase method (Beckman Coulter AU480 Chemistry Analyser, Johannesburg, South Africa), with sufficient sensitivity (range 0.555–44.4 mol/L) to determine glucose concentration in the study samples. Serum total cholesterol, triglyceride, low-density lipoprotein cholesterol (LDL-C) and high-density lipoprotein cholesterol (HDL-C) were measured using enzymatic assays on the Beckman Coulter Chemistry Analyser. The same protocol was used to take these measures at follow-up.

**Outcome.** The participants were classified as having diabetes at follow-up if: the participant had since received a diagnosis of diabetes by health personnel, they were currently on diabetes medication, or the FPG at follow up was greater than or equal to 7.0 mmol/L. IFG was diagnosed if FPG was greater or equal to 6.0 mmol/L and less than 7.0 mmol/L. Normal Glucose Tolerance (NGT) individuals were those with FPG less than 6mmol/L.

**Statistical analysis.** Statistical analyses were performed using Stata version 16 (StataCorp LP, College Station, TX). The baseline characteristics are expressed as mean ± SD or median (interquartile range [IQR]) for continuous variables or proportions for categorical variables. Person-years for diabetes were calculated from baseline until the last examination. Incidence of diabetes with 95% confidence interval (CI) was calculated per 1000 person-years, using the number of participants who developed diabetes during follow-up as the numerator and the

total person-time as the denominator. We only had diagnosis information for 29% of people who developed diabetes. The remaining 71% were identified during follow-up. Therefore, we don't know the exact time it took for them to progress from the starting point. Because of this, we used logistic regression instead of survival analysis to analyze our data.

Variables known or suspected to attribute to the risk of progression were included in univariate analysis and then in multivariable logistic models adjusting for three important covariates: age, BMI and waist circumference. All models were adjusted for the duration of follow-up. The covariates included were age, BMI, waist circumference, waist-hip ratio, systolic blood pressure, baseline FPG, cholesterol, triglycerides, HDL-C and LDL-C, sex, site, wealth score, education, smoking history and HIV status. Longitudinal changes of risk factors in those that progressed to diabetes, remit to normal glucose regulation, or remained IFG, were calculated and one-way ANOVA was used to compare the three progression groups.

Models of risks associated with the development of diabetes within four years using simple measures routinely taken in the clinical care performed were developed with performance determined by pseudo $R^2$ and area under the curve (AUC). We provide a risk classification tool with probabilities of developing diabetes, using the most practical model to guide clinical interventions.

Sensitivity analysis was performed to assess any differences between those who were followed up and those who could not be found at follow up.

## Results

The follow-up study was conducted in 2018–2019 with a median of 4.2 years (714.9 person-years to follow-up). A total of 190 participants, of the 389 who had IFG in the population, were successfully traced during the follow-up period representing 49% identification. Fifteen of those traced were confirmed to have died, and we report the results of 175 participants (Fig 1). Baseline characteristics of those followed up and lost to follow up are shown in S1 Table. The participants successfully followed-up were more likely to be rural residents and older than those not followed-up. There were no differences in the sex or baseline glucose results between the two populations.

Forty-five (26%) participants with IFG at baseline progressed to diabetes (incidence rate of 62.9 per 1000 person-years 95% CI 47.0–84.3). Of those who developed diabetes, the majority, 32 (71%), did not know their status and were detected during the follow-up. 106 (61%) participants regressed to normal glucose tolerance, and 24 (14%) remained as IFG at follow up Table 1 shows the clinical, biochemical and demographic characteristics of the participants at baseline based on their glycaemic status at follow-up. The individuals who progressed from IFG were older, and had higher BMI, waist circumference, systolic and diastolic BP, FPG, cholesterol, triglycerides and LDL-C compared to those who did not. There were no significant differences in changes of characteristics between those who remitted to normal glucose tolerance remained IFG or progressed to diabetes in the study period, apart from the change in age, which reflects the differences in the follow-up period (S2 Table).

Table 2 shows the multivariable logistic regression analysis representing the odds ratio (OR) and 95% CI for progression to diabetes. Standardised logistic regression adjusted for follow-up for each variable showed the strongest predictors of progression were BMI (OR 1.98, 95% CI 1.34–2.94, $P = 0.001$), waist circumference, (OR 2.50, 95% CI 1.60–3.91, $P < 0.001$) and LDL-C (OR 1.80, 95% 1.23–2.64, $P = 0.002$). Higher waist-hip ratio (OR 1.40, 95%CI 0.98–2.01, $P = 0.03$), systolic blood pressure (OR 1.56, 95% CI 1.10–2.21, $P = 0.01$) and FPG (OR 1.53, 95%CI 1.08–2.16, $P = 0.01$), were also associated with progression.

In models adjusted for both age and follow-up, BMI and waist circumference remained strong predictors; waist-hip ratio, FPG and LDL-C remained associated with progression, but systolic blood pressure was not. However, in models adjusted for waist circumference and

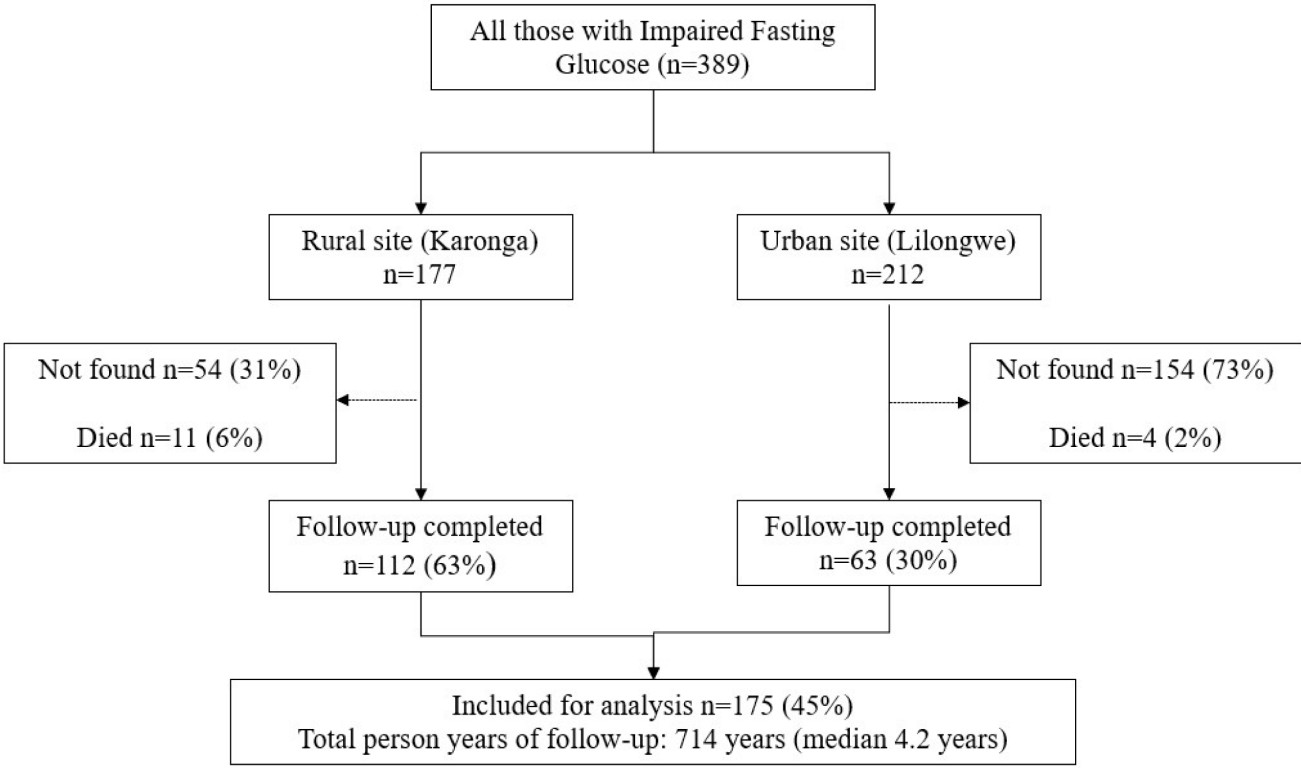

**Fig 1. Flow diagram of study participants by site.**

follow-up period, only FPG was associated with progression. Among the measures of adiposity, waist circumference was the biggest predictor of progression to diabetes in the population. Analysis using participants only from the rural area which had a greater follow-up rate produced similar results (S3 Table).

Models of risks associated with progression to diabetes using simple measures routinely taken in clinical care (age, BMI, waist circumference and FPG) are presented in Table 3. The AUC values of models of follow-up period with age, BMI and waist circumference were 0.69, 0.75 and 0.77, respectively. In models adjusted for follow-up, a model with two covariates, waist circumference and FPG, had the higher pseudo $R^2$ and AUC (0.186 and 0.79); adding age to this model only marginally improved these measures. A graph of probabilities of developing diabetes over four years based on the model of follow-up period, waist circumference and FPG is represented in Fig 2.

## Discussion

This study is among the first to determine the incidence of diabetes in participants with IFG, as defined by WHO classification, in a rural and urban population in Malawi. We found a high incidence of diabetes, with 26% of participants progressing in 4.2 years. Waist circumference and baseline FPG were the two most significant predictors, and a model of these risk factors can potentially be used as a tool in screening for those at risk of progression to diabetes.

### Comparison with previous research

The incidence rate of diabetes in our cohort is 62.9 per 1000 person-years, which is higher than previously reported in studies of white Europeans (35.0–40.0 per 1,000 person-years) and Iran (34.5 per 1,000 person-years) [19] but similar to that reported in Chennai India (61.0 per

**Table 1. Baseline characteristics of participants with IFG (n = 175) based on their glycaemic status at follow-up.**

| Variable | n | Regressed to NGT 106(60.6) | Remained as IFG 24(13.7) | Progressed to DM 45(25.7) |
|---|---|---|---|---|
| Location: Rural, n (%) | 112 (64) | 75(67) | 13(11.6) | 24(21.4) |
| Urban, n (%) | 63 (36) | 31(49.2) | 11(17.5) | 21(33.3) |
| Sex: Male, n (%) | 54 (30.8) | 35(64.8) | 6(11.1) | 13(24.1) |
| Female, n (%) | 121(69.1) | 71(58.7) | 18(14.9) | 32(26.4) |
| Age, (years) | 175 | 44.1±14.5 | 57.4±12.5 | 52.4±13.1 |
| BMI, (kg/m$^2$) | 173 | 25.7±5 | 29±6.3 | 30.9±6 |
| Waist Circumference, (cm) | 173 | 87±10.7 | 94±10.5 | 99±11.2 |
| Waist Hip Ratio | 173 | 0.9±0.1 | 0.9±0.1 | 0.9±0.1 |
| Systolic BP, (mmHg) | 175 | 131±22 | 144±25 | 146±28 |
| Fasting plasma glucose, (mmol/L) | 175 | 6.3±0.2 | 6.4±0.2 | 6.5±0.3 |
| Serum Cholesterol, (mmol/L) | 175 | 4.5±1.2 | 4.9±0.9 | 5.1±1.1 |
| Serum Tryglycerides, (mmol/L) | 175 | 1.3±0.8 | 1.9±1.2 | 1.9±1 |
| HDL Cholesterol, (mmol/L) | 173 | 1.1±0.4 | 1.1±0.2 | 1.1±0.2 |
| LDL Cholesterol, (mmol/L) | 175 | 2.9±0.9 | 3.4±0.8 | 3.6±0.8 |
| HIV Status: Positive, n (%) | 15 | 13(86.7) | 1(6.7) | 1(6.7) |
| Positive family history of diabetes, n (%) | 26 (14.8) | 13(50) | 5(19.2) | 8(30.8) |
| Current consumption of Alcohol, n (%) | 27(15.4) | 20(74.1) | 2(7.4) | 5(18.5) |
| Wealth score: Poorest, n (%) | 40 (22.8) | 29(72.5) | 6(15) | 5(12.5) |
| Middle, n (%) | 75 (42.9) | 46(61.3) | 8(10.7) | 21(28) |
| Wealthiest, n (%) | 60 (34.3) | 31(51.7) | 10(16.7) | 19(31.7) |
| Physical activity: High (%) | 154 (88) | 95(89.6) | 20(83.3) | 39(86.7) |
| Moderate (%) | 15 (0.09) | 8(7.6) | 4(16.7) | 3(6.7) |
| Low (%) | 6 (0.03) | 3(2.8) | 0 | 3(6.7) |

Values are presented as mean ± SD for continuous variables and n(%) for proportions. NGT, Normal Glucose Tolerance; IFG, Impaired Fasting Glucose; BMI, body mass index; HDL, high-density lipoprotein; LDL, low-density lipoprotein.

1,000 person-years) [20]. The incidence is also similar to that reported in ethnic groups with high levels of obesity and diabetes, such as the Pima Indians (87.3 per 1,000 person-years), the Micronesian population of Nauru (62.8 per 1,000 person-years) and the Native Americans (66.1 per 1,000 person-years) [21].

Our data shows that the risk of progression to diabetes significantly increased as FPG and waist circumference increased. Several studies have demonstrated that the progression from prediabetes to diabetes is accompanied by worsening weight gain, insulin resistance, and beta-cell dysfunction [22]. Waist circumference is an inexpensive marker of insulin resistance and visceral adipose tissue [23]. Individuals with high waist circumference would be ideal candidates for intervention programs aimed at screening and preventing diabetes, which can potentially decrease the high rate of undiagnosed diabetes found in our study and others in the region. Lifestyle interventions, including either physical activity alone or combined with dietary advice aiming at weight loss, have been shown to significantly reduce the progression of prediabetes to diabetes in other populations [24–26]. However, these prevention programs have shown variable levels of success in some ethnicities, and it is not known if they can have an impact on SSA populations. Determining whether waist circumference reduction through a trial of dietary and physical activity to reduce the incidence of diabetes in rural and urban Malawi would be an appropriate next step.

Blood pressure and triglycerides are highly correlated with obesity [27]. Several studies also found that higher triglycerides and blood pressure at baseline in those with prediabetes were

**Table 2. Multivariate logistic regression of risk factors for progression of impaired fasting glucose (IFG) to diabetes.**

| Variables | Model 1: Standardised logistic regression adjusted for follow-up | | Model 2: Adjusted for follow-up and age | | Model 3: Adjusted for follow-up and BMI | | Model 4: Adjusted for follow-up and waist circumference | |
|---|---|---|---|---|---|---|---|---|
| | OR (95% CI) | P Value | OR (95% CI) | P Value | OR (95% CI) | P Value | OR (95% CI) | P Value |
| Age | 1.48 (1.01–2.19) | 0.046 | - | | 1.41 (0.93–2.13) | 0.11 | 1.27 (0.82–1.95) | 0.28 |
| BMI | 1.98 (1.34–2.94) | 0.001 | 1.95 (1.31–2.91) | 0.001 | - | | 1.01 (0.53–1.93) | 0.98 |
| Waist circumference | 2.50 (1.60–3.91) | <0.001 | 2.39 (1.52–3.76) | <0.001 | 2.48 (1.19–5.16) | 0.02 | - | |
| Waist hip ratio | 1.40 (0.98–2.01) | 0.03 | 1.30 (0.90–1.88) | 0.16 | 1.47 (1.01–2.14) | 0.046 | 1.05 (0.70–1.56) | 0.82 |
| Systolic BP | 1.56 (1.10–2.21) | 0.01 | 1.43 (0.97–2.11) | 0.07 | 1.40 (0.98–2.01) | 0.06 | 1.30 (0.90–1.87) | 0.16 |
| Fasting plasma glucose | 1.53 (1.08–2.16) | 0.01 | 1.45 (1.02–2.06) | 0.04 | 1.54 (1.07–2.22) | 0.02 | 1.55 (1.07–2.26) | 0.02 |
| Cholesterol | 1.44 (1.00–2.08) | 0.05 | 1.31 (0.88–1.94) | 0.19 | 1.23 (0.83–1.82) | 0.30 | 1.18 (0.79–1.76) | 0.42 |
| Triglycerides | 1.36 (0.95–1.94) | 0.09 | 1.28 (0.89–1.83) | 0.18 | 1.22 (0.84–1.77) | 0.31 | 1.07 (0.72–1.57) | 0.75 |
| HDL- Cholesterol | 1.14 (0.80–1.61) | 0.47 | 1.10 (0.77–1.57) | 0.58 | 1.21 (0.83–1.77) | 0.32 | 1.31 (0.88–1.94) | 0.18 |
| LDL- Cholesterol | 1.80 (1.23–2.64) | 0.002 | 1.69 (1.14–2.52) | 0.01 | 1.52 (1.01–2.27) | 0.04 | 1.45 (0.96–2.17) | 0.08 |
| Site | 1.36 (0.64–2.85) | 0.41 | 1.53 (0.71–3.27) | 0.27 | 0.93 (0.41–2.07) | 0.85 | 1.16 (0.53–2.54) | 0.72 |
| Sex | 1.09 (0.50–2.35) | 0.84 | 1.12 (0.51–2.46) | 0.77 | 0.72 (0.31–1.66) | 0.44 | 0.97 (0.42–2.21) | 0.93 |
| Wealth | 1.40 (0.86–2.28) | 0.18 | 1.44 (0.88–2.36) | 0.15 | 1.09 (0.64–1.85) | 0.76 | 0.99 (0.57–1.71) | 0.97 |
| Physical activity | 0.88 (0.43–1.81) | 0.72 | 0.96 (0.46–2.01) | 0.92 | 0.83 (0.36–1.83) | 0.64 | 0.91 (0.40–2.07) | 0.83 |

BMI, body mass index; HDL, high-density lipoprotein; LDL, low-density lipoprotein.

associated with progression to diabetes [28]. However, we found that the relationship apparent in univariate analysis was lost when adjusted for BMI or waist circumference. We observed that the association between age and incident diabetes was weak and no longer significant after adjustment for BMI or waist circumference. This was similar to what was observed in the Diabetes Prevention Program trial which demonstrated that intensive lifestyle intervention, to a lesser extent, metformin use reduced the risk of diabetes progression in high-risk adults [5, 8].

We did not observe a significant difference in the incidence of diabetes between men and women. This is contrary to several studies that have reported a higher incidence in women than men usually ascribed to higher adiposity and less physical activity [20]. Our baseline survey identified self-reported physical activity levels well above the WHO recommendation for men and women [15]. Contrary to other reported studies, we did not find that family history was a significant risk factor for progression to diabetes in people with IFG. This might be because some family members were unaware that they had diabetes, did not survive long

**Table 3. Models of risks associated with progression to diabetes.**

| Model No: | Variables | N | Pseudo R2 | AUC | 95% CI |
|---|---|---|---|---|---|
| 1 | Follow up period, Age | 175 | 0.0831 | 0.695 | 0.61–0.78 |
| 2 | Follow up period, Fasting glucose | 175 | 0.0910 | 0.702 | 0.62–0.79 |
| 3 | Follow up period, BMI | 173 | 0.1284 | 0.754 | 0.67–0.84 |
| 4 | Follow up period, Waist | 173 | 0.1590 | 0.767 | 0.69–0.85 |
| 5 | Follow up period, Waist, Age | 173 | 0.1648 | 0.77 | 0.69–0.85 |
| 6 | Follow up period, Waist, BMI | 173 | 0.1650 | 0.767 | 0.69–0.84 |
| 7 | Follow up period, Waist, BMI, Age | 173 | 0.1650 | 0.775 | 0.69–0.85 |
| 8 | Follow up period, Waist, Fasting glucose | 173 | 0.1861 | 0.786 | 0.71–0.86 |
| 9 | Follow up period, Waist, Fasting glucose, Age | 173 | 0.1881 | 0.792 | 0.72–0.87 |

AUC, area under the receiver operator characteristics (ROC) curve; CI confidence interval

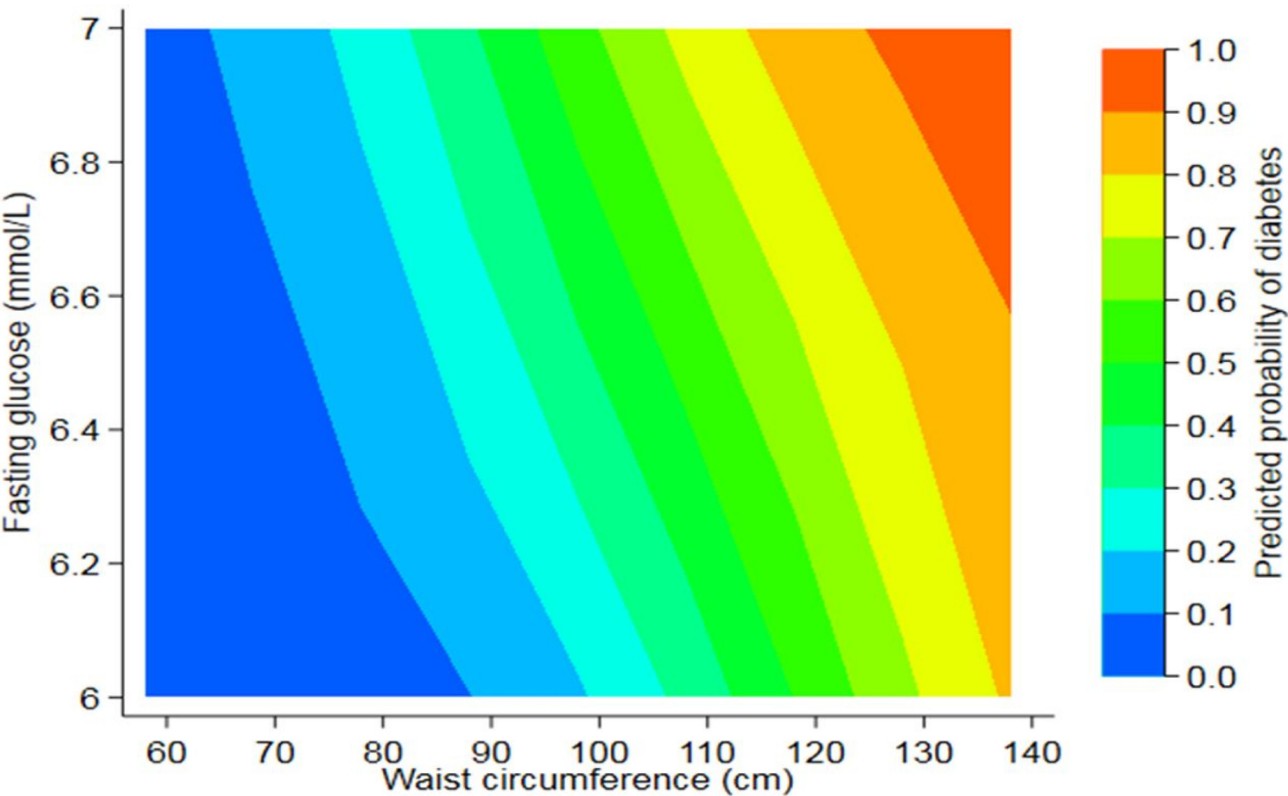

**Fig 2. Probabilities of progression to diabetes within four years based on a simple regression model of waist circumference and fasting glucose over four years.** Predicted probability of 1.0 represents 100% risk of developing diabetes.

enough to develop diabetes, or were not exposed to the same precipitating factors as their index relatives.

### Implications and future directions

Strengths of the study include that the study recruited participants from both urban and rural populations. The participants were visited at home, early in the morning, so there is no effect of physical activity on glucose results. Furthermore, all the tests were done to a standard protocol with the same equipment.

A large proportion of individuals with prediabetes at baseline regressed to normal glucose regulation at follow-up. This has also been observed in most cohort studies in different regions and questions the usefulness of surveys as a tool for diabetes prevention [29–31]. It is possible that many of the populations may pass in and out of prediabetes without a diagnosis, and many people found in surveys could well have transient diabetes. This places greater importance on identifying those individuals at greatest risk of developing diabetes.

Our data suggest that a simple contour map with probabilities of progression to diabetes based on waist circumference and FPG could be used in a clinical setting to identify those at risk of progression (Fig 2). Glucose tests are now available worldwide, so this map is a practical and pragmatic tool that health officials could use to communicate the likelihood of patients with IFG to progress to diabetes using simple clinical features without expensive tests or age, which is not always accurate. Further tests are needed to validate this map in this population and other SSA populations.

## Limitations

The study's limitations are the lack of year-by-year follow-up data and the relatively low response rate, especially among men and urban participants. Loss to follow-up remains a significant challenge in conducting long term research and intervention studies in many countries in SSA. Our study had a better follow-up in the rural area (75%), and a repeat analysis using participants only from this region produced similar results. We recruited fewer participants than anticipated which means that the 95% confidence is broad, 47.0–84.3 per 1000 person-years. However, this is the first information on the progression from intermediate hyperglycaemia in Malawi and so it will inform future planning in the country and possibly other countries in sub-Saharan Africa.

Research needs to be done to understand how to improve retention of patients in studies in sub-Saharan Africa or follow up people at risk of diabetes or those with the disease. This could involve developing national registers or regular record collection of people with prediabetes or diabetes and qualitative work to understand the barriers and facilitators to retaining people in studies.

Another limitation of the study was that oral glucose tolerance tests were not conducted, meaning isolated impaired glucose tolerance could not be identified. However, several other studies in this population have also found that fasting glucose alone identified most individuals with prediabetes, suggesting that glucose tolerance testing would not be cost-effective [32].

## Conclusion

We found that the incidence rate of diabetes in people with IFG in Malawi is higher than that seen in Europe or the USA but similar to that seen in India. However, the predictors of progression to diabetes are similar, with waist circumference and glycaemia being the strongest predictors of progression. Based on modelling, we suggest that a simple chart with probabilities of progression to diabetes based on waist circumference and FPG could be developed for use in a clinical setting in sub-Saharan Africa to identify those at risk of progression.

## Supporting information

**S1 Table. Comparison between participants who were found and not found at follow up.**
(DOCX)

**S2 Table. Longitudinal changes of risk factors in those that remit to normal glucose tolerance (NGT) remained impaired fasting glucose (IFG) or progressed to diabetes.**
(DOCX)

**S3 Table. Multivariate logistic regression of risk factors to the progression of IFG using participants from the rural area.**
(DOCX)

**S4 Table. Diagnostic accuracy of waist circumference and baseline fasting glucose concentration as a screening tool for predicting the progression of diabetes in people with prediabetes.**
(DOCX)

## Acknowledgments

We acknowledge Dr Angus G Jones and Dr Beverly S Shields for their assistance in several stages of this study.

## Author Contributions

**Conceptualization:** Wisdom P. Nakanga, Amelia C. Crampin, Moffat J. Nyirenda.

**Formal analysis:** Andrew T. Hattersley, Lauren R. Rodgers.

**Funding acquisition:** Moffat J. Nyirenda.

**Investigation:** Wisdom P. Nakanga, Joseph Mkandawire, Louis Banda.

**Methodology:** Wisdom P. Nakanga, Amelia C. Crampin.

**Project administration:** Amelia C. Crampin.

**Supervision:** Amelia C. Crampin, Rob C. Andrews, Andrew T. Hattersley, Moffat J. Nyirenda.

**Visualization:** Wisdom P. Nakanga, Lauren R. Rodgers.

**Writing – original draft:** Wisdom P. Nakanga.

**Writing – review & editing:** Amelia C. Crampin, Joseph Mkandawire, Rob C. Andrews, Andrew T. Hattersley, Moffat J. Nyirenda, Lauren R. Rodgers.

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
