## [Decision Letter · Decision Letter 0]

19 Dec 2022

PGPH-D-22-01643

Waist circumference and glycaemia are strong predictors of progression to diabetes in individuals with prediabetes in sub-Saharan Africa: 4-year prospective cohort study in Malawi

Dear Dr. Nakanga,

Thank you for submitting your manuscript to PLOS Global Public Health. After careful consideration, we feel that it has merit but does not fully meet PLOS Global Public Health’s publication criteria as it currently stands. Therefore, we invite you to submit a revised version of the manuscript that addresses the points raised during the review process.

EDITOR:

This manuscript still requires major corrections; please attend to all the reviewers' comments.

The decision of this manuscript is justified based on PLOS Global Public Health’s publication criteria and not on its novelty or perceived impact.

We look forward to receiving your revised manuscript.

Kind regards,

Zulkarnain Jaafar

Academic Editor

Journal Requirements:

1. Please send a completed 'Competing Interests' statement, including any COIs declared by your co-authors. If you have no competing interests to declare, please state "The authors have declared that no competing interests exist". Otherwise please declare all competing interests beginning with the statement "I have read the journal's policy and the authors of this manuscript have the following competing interests:"

2. Please provide separate figure files in .tif or .eps format only and remove any figures embedded in your manuscript file. Please also ensure that all files are under our size limit of 10MB.

Additional Editor Comments (if provided):

Reviewers' comments:

Reviewer's Responses to Questions

**Comments to the Author**

1. Does this manuscript meet PLOS Global Public Health’s publication criteria? Is the manuscript technically sound, and do the data support the conclusions? The manuscript must describe methodologically and ethically rigorous research with conclusions that are appropriately drawn based on the data presented.

Reviewer #1: Yes

Reviewer #2: Yes

2. Has the statistical analysis been performed appropriately and rigorously?

Reviewer #1: Yes

Reviewer #2: Yes

3. Have the authors made all data underlying the findings in their manuscript fully available (please refer to the Data Availability Statement at the start of the manuscript PDF file)?

Reviewer #1: Yes

Reviewer #2: Yes

4. Is the manuscript presented in an intelligible fashion and written in standard English?

Reviewer #1: Yes

Reviewer #2: Yes

5. Review Comments to the Author

Reviewer #1: The authors used data from an extensive rural and urban non-communicable disease survey. One hundred seventy-five, of 389 individuals with impaired fasting glucose (IFG) at baseline, age 48 ±15 years and body mass index 27.5 ±5.9 kg/m2 were followed up for a median of 4.2 years (714 person-years). Incidence rates were calculated, and predictors of progression to diabetes were analysed using multivariable logistic regression models, with overall performance determined using receiver operator characteristics (ROC) curves.The median follow-up was 4.2 (IQR 3.4 – 4.7) years. Forty-five out of 175 (26%) progressed to diabetes. Incidence rates of diabetes were 62.9 per 1000 person-years 95% CI, 47.0-84.3. The predictors of progression were higher; age (odds ratio [OR] 1.48, 95% CI 1.01-2.19, P=0.046), BMI (OR 1.98, 95% CI 1.34-2.94, P=0.001), waist circumference (OR 2.50, 95% CI 1.60-3.91,P<0.001), waist-hip ratio (OR 1.40, 95%CI 0.98-2.01, P=0.03), systolic blood pressure (OR 1.56, 95% CI 1.10-2.21, P=0.01), fasting plasma glucose (OR 1.53, 95%CI 1.08-2.16, P=0.01), cholesterol (OR 1.44, 95% CI 1.00-2.08, P=0.05) and low-density lipoprotein cholesterol (OR 1.80, 95% 1.23-2.64, P=0.002). A simple model combining fasting plasma glucose and waist circumference was predictive of progression to diabetes (ROC area under the curve=0.79) The incidence of diabetes in people with IFG in Malawi is higher than those seen in Europe (35.0 per 1,000 person-years) but similar to those seen in India (61.0 per 1,000 person-years).

(Put inserts in a different color font to identify changes in the article)

Include in article

1 - Abstract

Conclusions: State only what your study found; do not include extraneous information not backed up by the results.

2 - Discussion

Compare and contrast your study with others in the most relevant world literature, particularly the recent literature.

3 - What new information is sufficient to modify existing clinical practice?

4 -What are the conclusions and implications for current practice, and particularly for future research that may have a significant impact on clinical decisions?

5 - How can this study affect public policies related to health?

6 - What does this study add to the literature?

7 - At the end of the Discussion, under the subheading "Limitations," review the limitations of your study.

8 - At the end of the limitations, under the subheading " Future directions".

9 - Conclusion

Take special care to draw your conclusions only from your results and verify that your conclusions are firmly supported by your data

10 - Table 1

I suggest include: metabolic syndrome, lipids ratio, exercise, hypertension, diabetes, smokers, liver steatosis, alcohol intake, Non-HDL cholesterol, COPD¸ Gout, Carotid artery disease, Hypothyroidism, Anemias, Drug abuse, asthma, atrial fibrillation, high-sensitivity C-reactive protein, Chronic kidney disease, Dialysis, Heart failure, Liver cirrhosis, Gastrointestinal bleeding history, Fasting glucose, Depression, Valve disorder, Blood pressure (Systolic and Diastolic), Heart rate, bpm; Hemoglobin, g/dL, mean (SD), Platelet, x 109/L, mean (SD) and medications. Family history of stroke, CAD, PAD. History of cancer.

11– References

Update

Include a list of abbreviations under all tables.

Include a list of abbreviations before the Abstract.

Improve the visualization quality of figures.

12 - How was the sample size calculated?

13 - Why was it made to use statistical tests?

14- Include a table with measures of sensitivity, specificity, positive and negative predictive values, positive and negative likelihood ratios, area under the ROC curve, diagnostic odds and Youden index.

Reviewer #2: Authors tried to identify the predictors for diabetes by conducting a prospective cohort study.

However the identified predictors are waist circumstances and glycemia are the predictors and this is not a new information. However, authors attempted to find these in sub-sehran africa area. Where information is very limited. Few of my comments are as mentioned below:

Table 1 and 2 may be given with significant values in bold.

Cite a reference for the article "Blood pressure and triglycerides are highly correlated with obesity. Several studies also

found that higher triglycerides and blood pressure at baseline in those with prediabetes were associated with progression to diabetes"

Authors unpublished observation can't be cited.

While submitting authors may insert line numbers.

6. PLOS authors have the option to publish the peer review history of their article (what does this mean?). If published, this will include your full peer review and any attached files.

**Do you want your identity to be public for this peer review?** For information about this choice, including consent withdrawal, please see our Privacy Policy.

Reviewer #1: No

Reviewer #2: **Yes: **Ramu Adela

---

## [Decision Letter · Decision Letter 1]

3 Mar 2023

PGPH-D-22-01643R1

Waist circumference and glycaemia are strong predictors of progression to diabetes in individuals with prediabetes in sub-Saharan Africa: 4-year prospective cohort study in Malawi

Dear Dr. Nakanga,

Thank you for submitting your manuscript to PLOS Global Public Health. After careful consideration, we feel that it has merit but does not fully meet PLOS Global Public Health’s publication criteria as it currently stands. Therefore, we invite you to submit a revised version of the manuscript that addresses the points raised during the review process.

EDITOR:

Dear Author,

This manuscript still requires some minor changes to be made. Please refer to the suggestions made by the reviewer.

The decision of this manuscript is justified based on PLOS Global Public Health’s publication criteria and not on its novelty or perceived impact.

We look forward to receiving your revised manuscript.

Kind regards,

Zulkarnain Jaafar

Academic Editor

Journal Requirements:

Additional Editor Comments (if provided):

Reviewers' comments:

Reviewer's Responses to Questions

**Comments to the Author**

1. If the authors have adequately addressed your comments raised in a previous round of review and you feel that this manuscript is now acceptable for publication, you may indicate that here to bypass the “Comments to the Author” section, enter your conflict of interest statement in the “Confidential to Editor” section, and submit your "Accept" recommendation.

Reviewer #1: (No Response)

2. Does this manuscript meet PLOS Global Public Health’s publication criteria? Is the manuscript technically sound, and do the data support the conclusions? The manuscript must describe methodologically and ethically rigorous research with conclusions that are appropriately drawn based on the data presented.

Reviewer #1: (No Response)

3. Has the statistical analysis been performed appropriately and rigorously?

Reviewer #1: (No Response)

4. Have the authors made all data underlying the findings in their manuscript fully available (please refer to the Data Availability Statement at the start of the manuscript PDF file)?

Reviewer #1: (No Response)

5. Is the manuscript presented in an intelligible fashion and written in standard English?

Reviewer #1: (No Response)

6. Review Comments to the Author

Reviewer #1: Authors do not adequately respond to comments.

Put inserts in a different color font to identify changes in the article)

Include in article

1 - Include a table with measures of sensitivity, specificity, positive and negative predictive values, positive and negative likelihood ratios, area under the ROC curve, diagnostic odds and Youden index.

2 - The subtopic Strengths can be added to Implications and future directions

7. PLOS authors have the option to publish the peer review history of their article (what does this mean?). If published, this will include your full peer review and any attached files.

**Do you want your identity to be public for this peer review?** For information about this choice, including consent withdrawal, please see our Privacy Policy.

Reviewer #1: No

---

## [Decision Letter · Decision Letter 2]

2 Jun 2023

PGPH-D-22-01643R2

Waist circumference and glycaemia are strong predictors of progression to diabetes in individuals with prediabetes in sub-Saharan Africa: 4-year prospective cohort study in Malawi

Dear Dr. Nakanga,

Thank you for submitting your manuscript to PLOS Global Public Health. After careful consideration, we feel that it has merit but does not fully meet PLOS Global Public Health’s publication criteria as it currently stands. Therefore, we invite you to submit a revised version of the manuscript that addresses the points raised during the review process.

EDITOR: 

Dear Author,

Please attend to all the comments provided by the reviewer/s and make necessary changes.

The decision of this manuscript is justified based on PLOS Global Public Health’s publication criteria and not on its novelty or perceived impact.

We look forward to receiving your revised manuscript.

Kind regards,

Zulkarnain Jaafar

Academic Editor

Journal Requirements:

Additional Editor Comments (if provided):

Reviewers' comments:

Reviewer's Responses to Questions

**Comments to the Author**

1. If the authors have adequately addressed your comments raised in a previous round of review and you feel that this manuscript is now acceptable for publication, you may indicate that here to bypass the “Comments to the Author” section, enter your conflict of interest statement in the “Confidential to Editor” section, and submit your "Accept" recommendation.

Reviewer #3: (No Response)

2. Does this manuscript meet PLOS Global Public Health’s publication criteria? Is the manuscript technically sound, and do the data support the conclusions? The manuscript must describe methodologically and ethically rigorous research with conclusions that are appropriately drawn based on the data presented.

Reviewer #3: Partly

3. Has the statistical analysis been performed appropriately and rigorously?

Reviewer #3: I don't know

4. Have the authors made all data underlying the findings in their manuscript fully available (please refer to the Data Availability Statement at the start of the manuscript PDF file)?

Reviewer #3: Yes

5. Is the manuscript presented in an intelligible fashion and written in standard English?

Reviewer #3: No

6. Review Comments to the Author

Reviewer #3: Title: Waist circumference and glycaemia are strong predictors of progression to diabetes in individuals with prediabetes in subSaharan Africa: 4-year prospective cohort study in Malawi

Comment: the title is too long, it also gives an implication about the findings of the study not the objective. We can not say “Waist circumference and glycaemia are strong predictors of progression to diabetes” This means we already know the answer to our research question in advance. Also, the title includes two settings “Sub-Saharan African” and “Malawi”. The two are different and cannot represent each other.

Authors need to change the title.

Abstract

Results:

• “The predictors of progression were higher; age (odds ratio [OR] 1.48, 95% CI 1.01-2.19, P=0.046), BMI (OR 1.98, 95% CI 1.34-2.94, P=0.001), waist circumference (OR 2.50, 95% CI 1.60-3.91……..”

Comment: I find this sentence too long and unclear to me.

Conclusions:

• ………………and in predictors of progression are like those seen in other populations”

Comment: This sentence needs to be rephrased

• A simple chart with probabilities of progression to diabetes based on waist circumference and fasting plasma glucose could be used to identify those at risk of progression in clinical settings in sub-Saharan Africa.

Comment: This is not a conclusion. It could be a recommendation.

Introduction:

In this section, the authors provided on information on diabetes in Malawi per se, the country where study was conducted. All information was on SSA and this could not be representative of the country.

Also, the last paragraph on diagnosis of prediabetes is too irrelevant. Authors would better provide definition of prediabetes and its prevalence in similar contexts and different age groups.

Methods:

• “We used the Cochran formula to calculate the power of the study.”

Comment: It is unclear to me what authors meant by the sentence.

• “We combined self-reported physical exercise duration (minutes) and intensity (pre-coded activities, grouped into high exertion, low exertion, or sedentary) in the previous week (both at work and during leisure time) to generate average metabolic equivalent of task (MET) data per day, and categorised these into whether or not the WHO recommendations of at least 600 Total Physical Activity MET minutes per week were met”.

Comment: I am not sure how combining the duration and intensity of physical exercise can generate the MET? Is there a reference for that?

Also this part: categorised these into whether or not the WHO recommendations of at least 600 Total Physical Activity MET minutes per week were met” is totally unclear

• ……..a trained nurse who also provided HIV screening using a rapid diagnostic test

Comment: In the Biochemical analysis it is not clear to me why was HIV screening done?

• ……..the participant had since received a diagnosis of diabetes by health personnel”

Comment: Since when?

Statistical analysis

“We only had diagnosis data on 29% of those who progressed to diabetes. 71% were identified at follow-up. Thus exact time to progression from baseline is unknown. Therefore, we used logistic regression rather than survival analysis to interpret our data.”

Comment: This sentence is unclear to me. Please rephrase for better explanation.

Results:

• “There were no significant differences in changes of clinical characteristics between those who remitted to normal glucose tolerance remained IFG or progressed to diabetes in the study period, apart from the change in age…………”

Comment: What do authors mean by clinical characteristics? Especially age is not a clinical feature it is a demographic characteristic.

• In models adjusted for both age and follow-up……..

Comment: why were the models adjusted for follow up period? Was it not the same for all participants? And if not why?

Discussion:

• In this section the authors are assuming that their study findings can be generalized to SSA inhabitants while the truth is their sample is not even representative of Malawi, one country in the region given that sample is withdrawn from one rural and one urban settings. This is actually a limitation in the study.

• Also in the discussion, authors are in general repeating the findings in the “Results” section without providing any possible explanation for them. For example, they mentioned that unlike what is known in evidence: “we found that the relationship apparent in univariate analysis was lost when adjusted for BMI or waist circumference. In contrast to most studies, we observed that the association between age and incident diabetes was weak and no longer significant after adjustment for BMI or waist circumference”.

I would have expected them to try to find an explanation for that especially that in the introduction section they indicated one of the reasons for the study is that current evidence on factors for progression of prediabetes to diabetes in other countries can not be generalized to SSA.

• The Diabetes Prevention Program trial demonstrated that intensive lifestyle intervention, to a lesser extent, metformin use reduced the risk of diabetes progression in high-risk adults (5, 31).

Comment: I am not sure what is the relevance of those few lines where they are placed.

• Strengths of the study cannot fall under the “ Implications and future directions” section

• In the limitations section, the authors mentioned that :

“However, this is the first information on the progression from intermediate hyperglycaemia in sub-Saharan Africa and so it will inform future planning in the region”.

Comment: I totally disagree with this conclusion about informing the whole region

• “Our data suggest that a simple contour map with probabilities of progression to diabetes based on waist circumference and FPG could be used in a clinical setting to identify those at risk of progression”

Comment: I am not convinced that this map would be effective if based on investigations

in a clinical setting. How about people who do not visit clinics or live in hard-to-reach

areas?

All in all, the discussion section is messed up and needs to be refurbished and more evidence needs to be added to explain study findings. Recommendation that can overcome the study limitations should be included to guide future research.

7. PLOS authors have the option to publish the peer review history of their article (what does this mean?). If published, this will include your full peer review and any attached files.

**Do you want your identity to be public for this peer review?** For information about this choice, including consent withdrawal, please see our Privacy Policy.

Reviewer #3: No

---

## [Editor Report · Decision Letter 3]

6 Sep 2023

Waist circumference and glycaemia are strong predictors of progression to diabetes in individuals with prediabetes in sub-Saharan Africa: 4-year prospective cohort study in Malawi

PGPH-D-22-01643R3

Dear Dr Nakanga,

We are pleased to inform you that your manuscript 'Waist circumference and glycaemia are strong predictors of progression to diabetes in individuals with prediabetes in sub-Saharan Africa: 4-year prospective cohort study in Malawi' has been provisionally accepted for publication in PLOS Global Public Health.

Best regards,

Julia Robinson

Executive Editor